# Third Molar Agenesis Is Associated with Facial Size

**DOI:** 10.3390/biology10070650

**Published:** 2021-07-12

**Authors:** Nikolaos Gkantidis, Manuel Tacchi, Elias S. Oeschger, Demetrios Halazonetis, Georgios Kanavakis

**Affiliations:** 1Department of Orthodontics and Dentofacial Orthopedics, University of Bern, CH-3010 Bern, Switzerland; Manuel.tacchi@zmk.unibe.ch (M.T.); elias.oeschger@zmk.unibe.ch (E.S.O.); 2Department of Orthodontics, School of Dentistry, National and Kapodistrian University of Athens, GR-11527 Athens, Greece; dhalaz@dent.uoa.gr; 3Department of Pediatric Oral Health and Orthodontics, UZB—University School of Dental Medicine, University of Basel, CH-4058 Basel, Switzerland; 4Department of Orthodontics, Tufts University School of Dental Medicine, Boston, MA 02111, USA

**Keywords:** growth/development, tooth agenesis, morphogenesis, morphometrics, tooth development, wisdom tooth, third molar

## Abstract

**Simple Summary:**

Missing third molars is a common occurrence in modern humans with a prevalence of approximately 20% in the general population. The absence of those teeth, however, is not found in other human predecessors. Therefore, there is speculation whether the congenital absence of third molars is part of an evolutionary mechanism that leads to smaller jaws, smaller and fewer teeth, or if their absence is associated with more local developmental factors. In this study, we assessed the size of the cranial base, the maxilla, the mandible and the entire craniofacial complex in individuals missing one or more third molars and compared them with a group with no missing teeth. We showed that in cases with one or more missing third molars, there is a significant decrease in the size of the maxilla, the mandible as well as the entire facial configuration. Additionally, the more missing third molars, the smaller the jaws and the face were. These findings suggest that isolated third molar agenesis is part of a developmental mechanism related to craniofacial size reduction. Whether this mechanism is part of an evolutionary process in humans remains to be seen.

**Abstract:**

Individuals with congenitally missing permanent teeth, other than third molars, present smaller craniofacial configurations compared to normal controls. However, it is not known if agenesis of third molars is part of the same mechanism. Therefore, this study assessed individuals with and without isolated third molar agenesis and tested the relation of this condition to the size of their facial configurations, using geometric morphometric methods. We show that the absence of one or more third molars is associated with a smaller maxilla, smaller mandible and a smaller overall facial configuration. The effect was larger as the number of missing third molars increased. For example, the size of the mandibular centroids in five 16-year-old females with no, one, two, three or four missing third molars showed a size reduction of approximately 2.5 mm per missing third molar. In addition, in cases with third molar agenesis in one jaw only, the effect was also evident on the opposite jaw. Our findings suggest that isolated third molar agenesis is part of a developmental mechanism resulting also in craniofacial size reduction. This might be the effect of an evolutionary process observed in humans, leading to fewer and smaller teeth, as well as smaller facial structures.

## 1. Introduction

A tooth agenesis is described as the developmental absence of one or more teeth. The origin of tooth agenesis is genetic; it is often a phenotypic trait of craniofacial syndromes, but can also present itself as a separate entity, namely as non-syndromic tooth agenesis. In the permanent dentition, non-syndromic tooth agenesis in teeth other than third molars has a prevalence of 6.4% and occurs more often in females than males [1,2]. However, the absence of third molars alone is much more prevalent, occurring in approximately 20–30% of the population [2,3]. Therefore, in cases where all other permanent teeth are present, an agenesis of the third molars is largely viewed as a normal phenotype. Nevertheless, previous studies have shown that there is a strong association between missing third molars and agenesis of other teeth, with the latter increasing the chance of third molar agenesis by approximately 40% [4,5,6]. This raises the question, whether the presence of third molars is controlled by the same genetic mechanisms as other permanent teeth, or their formation follows a separate developmental pattern.

In humans, there are several genes that have been linked to tooth agenesis [7,8], some of which are also involved in the morphogenesis of the craniofacial structures. From an evolutionary viewpoint, this might indicate a common mechanism for tooth and craniofacial formation. It is documented that during human evolution, the size of the brain, along with the cranium, has continuously increased [9] as a response to dietary or societal factors [10]. On the other hand, facial structures have become smaller, and the number of teeth has declined [11,12,13]. It has been speculated that a size reduction in the tooth-bearing structures would lead to a reduction in tooth size and tooth number, as supported by the fact that humans present a significantly higher incidence of tooth agenesis compared to other primates [14,15]. Recent investigations support this notion and have showed that modern humans with tooth agenesis also present smaller facial dimensions. Notably, as the number of missing teeth increases, the size of the facial configuration appears to decrease [16].

Despite it being a more hominin-specific phenomenon, it remains unknown whether third molar agenesis is part of an evolutionary mechanism or a finding related to the development of the jaws and the cranium. A reduction in facial height [17], shortening of the jaws [18] and decrease in tooth size [19,20] are phenotypically linked to the absence of third molars, thus, underpinning the significance of more local factors. Furthermore, there is evidence of a plastic response of the masticatory system to dietary habits with agricultural populations presenting reduced bony dimensions of the masticatory apparatus [21]. However, the effect of environmental factors on the structural variability of the human cranium is relatively small [21], therefore the role of an evolutionary mechanism cannot be discounted.

Due to the much higher prevalence of missing third molars, these are mostly disregarded in studies exploring tooth agenesis in humans. Their individual effect on facial shape has been previously studied, revealing a relationship, which, however, cannot be interpreted with certainty because of the small sample size [22]. Considering the limited information provided in current literature and based on previous data regarding the agenesis of other permanent teeth, the aim of the present study was to assess the effect of third molar agenesis on the size of the craniofacial structures in humans. The research hypothesis was that individuals with missing third molars present smaller facial configurations compared to individuals with a complete permanent dentition. The results will provide further insight regarding the role of third molars in the development of the human cranium, which might also be associated with evolutionary effects.

## 2. Materials and Methods

### 2.1. Ethical Approval

The ethical approval of this case-control observational study was provided by the Ethics Commission of the Canton of Bern, Switzerland (Project-ID: 2018-01340) and the Research Committee of the School of Dentistry, National and Kapodistrian University of Athens, Greece (Project-ID: 281, 2 September 2016). The methods were carried out in accordance with the relevant guidelines and regulations. The participants signed an informed consent to allow the use of their data in the study. The study is reported according to the STROBE guidelines [23].

### 2.2. Sample

The initial sample was selected from a previously studied database, which is described elsewhere [16]. To increase the robustness of the present methodology, the initial sample was supplemented with additional subjects. Patients’ records from the following sources were reviewed to select the final study population: (a) University of Bern, Switzerland, (b) National and Kapodistrian University of Athens, Greece, (c) two private practices in Athens and two in Thessaloniki, Greece, and (d) one private practice in Biel, Switzerland. Archived consecutive patient files obtained from January 2002 until the end of September 2020 were searched for the identification of eligible subjects. In order to be included in the study the following inclusion criteria needed to be met:-Age between 9 years and 50 years at the time of pre-treatment records. For patients younger than 12.5 years old, panoramic radiographs at older ages, higher than 12.5 years old, were checked to avoid misdiagnosis of late forming third molars-Agenesis of one or more third molars-European (White) ancestry-Lateral cephalometric radiographs in maximal intercuspation, of diagnostic quality, with a reference ruler at the mid-sagittal plane for magnification adjustment-Panoramic radiographs of diagnostic quality for the identification of the structures of interest

Subjects were not included in the study when the following criteria were present (exclusion criteria):-Presence of systemic diseases, syndromes, or any other conditions that affect craniofacial development, as reported in the subjects’ medical record-Agenesis of additional teeth, other than third molars-Presence of severe dental anomaly affecting tooth number, size, or form in any tooth except for third molars-History of previous intervention known to influence craniofacial morphology, such as orthodontic treatment, prior to cephalometric image acquisition

The final study sample comprised 470 individuals, who met the above criteria. 398 were retrieved from the pre-existing sample [16] and 72 were added for the purpose of this study. Of the 470 subjects, 310 had no agenesis, whereas 160 presented with a varying number of missing third molas. Details on sample composition are provided in Table 1 and the age distribution within the sample is presented in Appendix A.

### 2.3. Data Collection

Patient files were thoroughly reviewed at the place of sample collection, including the medical and dental history, the intraoral and extraoral photographs, and the radiographs. For each patient, data including place of sample collection, sex, date of birth, date of image acquisition, panoramic radiographs, cephalometric radiographs, and congenitally missing teeth were gathered. The radiographs of the individuals that were initially considered eligible, were further reviewed by two independent researchers (E.O. and M.T.) for the identification of missing teeth. Any disagreements were resolved by consensus between them and the first author. All data were recorded in an Excel sheet (Microsoft Excel^®^, Microsoft Corporation, Redmond, WA, USA). The patterns of third molar agenesis in the sample were recorded using the TAC system [24,25]. This system assigns a binary value for each tooth in a dentition, indicating its presence or absence. Thus, it provides a unique identification number for each individual, representing a specific tooth agenesis pattern.

### 2.4. Size Assessment

All lateral cephalometric images were uploaded on Viewbox 4 software (dHAL software, Kifissia, Greece) for digitization and further analyses, following scaling to real size. In order to obtain size information, predefined craniofacial structures were described with geometric morphometric methods [26]. The primary tested outcomes in this investigation were the sizes of the cranial base, the maxilla, the mandible and the entire facial configuration. These were, thus, described using 15 curves, 11 landmarks and 116 semi-landmarks (Figure 1). The curves represented the outlines of the configurations under investigation and were manually placed by a single experienced examiner (E.O.). The 11 anatomical or fixed landmarks represented points with clear anatomical description [27], such as the anterior and posterior nasal spines (ANS and PNS). The remaining 116 landmarks were placed automatically along the curves at predetermined equidistant positions and were treated as semi-landmarks, which do not have an anatomical interpretation and therefore, do not show correspondence between samples. In the next step, the landmark configurations of all subjects were used to create an average configuration of the entire sample. In order to minimize bending energy and decrease variation in the position of semi-landmarks, these were allowed to slide along their respective curves by using the average configuration as reference [27,28]. After sliding, a new average configuration was created serving as reference to perform a subsequent sliding of the new configurations. This iterative process was repeated three times until no detectable change was observed in the average configuration.

The resulting landmark configurations were superimposed using Procrustes Superimposition [29] in order to transform landmark coordinates (space coordinates) into shape coordinates, that describe the position of each landmark in shape space [30]. The arithmetic mean position of all landmarks in a single configuration represents the centroid of this configuration; and centroid size (CS), is the square root of the sum of squared distances of a set of landmarks from their centroid [31]. The size of the craniofacial configurations in this study was determined by the natural logarithm of the centroid size (ln(CS)) in order to guarantee that for isotropic landmark variation the distribution in size space is isotropic and to ensure data normality [26].

### 2.5. Statistical Analyses

#### 2.5.1. Method Error

Method error was evaluated through repeated digitization and CS measurement of 30 randomly selected radiographs. No systematic error was detected in any CS variable. The mean difference between repeated ln(CS) measurements was negligible (all points: −0.0014 ± 0.0047, cranial base: −0.0010 ± 0.0097, maxilla: 0.0010 ± 0.0062, mandible: 0.0020 ± 0.0054) as was the mean absolute difference (all points: 0.0034 ± 0.0037, cranial base: 0.0067 ± 0.0070, maxilla: 0.0047 ± 0.0041, mandible: 0.0048 ± 0.0031). 

#### 2.5.2. Main Hypotheses Testing

To assess the effect of missing third molars on the size of craniofacial structures and the entire facial configuration, multiple linear regression models were developed with size variables being the dependent variables (ln(CS) of the entire facial configuration, ln(CS) of the cranial base, ln(CS) of the maxilla, ln(CS) of the mandible). The predictor variables were sex, age and number of missing third molars. In addition to significance testing, tests of between-subjects’ effects and parameters estimates were also calculated. The statistical analysis was conducted with SPSS software (v.27.0, SPSS Inc., Chicago, IL, USA). In all cases, a two-sided significance test was carried out at an alpha level of 0.05.

## 3. Results

Age and sex appeared to have a significant effect on the size of the cranial base, the maxilla, the mandible and the entire facial configuration. Younger individuals and females presented overall smaller facial structures (Table 2; Appendix A). Missing third molars were significantly associated with the size of the maxilla, the mandible and entire facial configuration, but not the size of the cranial base (Table 2 and Table 3). Size variability for each craniofacial configuration organized by number of missing third molars is presented in Appendix A. On average, females and males with third molar agenesis presented 1.6% and 2.0% smaller facial configurations compared to their age- and sex-matched controls, who had all third molars. The differences between groups were more pronounced in the maxilla and the mandible, which were smaller by 3.0% and 3.3% in females and by 3.3% and 3.6% in males, respectively (Table 3). Furthermore, as the number of missing third molars increased, the effect became larger. More specifically, based on the regression model, the size of the mandible in five 16-year-old females with no, one, two, three or four missing third molars would be 207.47 mm, 205 mm, 202.55 mm, 200.14 mm, 197.75 mm, respectively. In this case, there would be a size reduction of approximately 2.5 mm per missing third molar (Table 2).

To address the main research question more comprehensively, the study sample was also divided into subgroups based on the location of missing third molars: no third molar agenesis (N = 310) and maxillary agenesis only (N = 37) or mandibular agenesis only (N = 49). As depicted in Table 4 and Table 5, individuals with isolated maxillary or mandibular agenesis of third molars also presented with a smaller mandible or maxilla, respectively.

## 4. Discussion

This study evaluated the effect of congenitally missing third molars on the size of the cranial base, the maxilla, the mandible and the entire facial configuration, with the means of geometric morphometrics. The rationale for testing this association stems from a previous investigation showing that individuals with non-syndromic tooth agenesis present smaller maxilla and smaller overall facial configuration compared to normal controls [16]. In support to this observation, here we show that individuals with one or more missing third molars also present a smaller maxilla, smaller mandible and a smaller overall facial configuration compared to individuals without missing third molars. Our finding adds significant information to the literature discussing the origin of third molar agenesis. 

Contrary to the agenesis of other permanent teeth, missing third molars are commonly considered as a “normal” phenotypic variation in modern humans and are therefore often disregarded in clinical diagnosis and in relevant research studies. Indeed, this may be somewhat justified since more than 50% of the population present with some kind of abnormality of the third molars [32,33]. Their partial or complete absence, in particular, may present in more than 40% in some populations [34], making it a very common occurrence [3]. Missing third molars are not observed in other hominins or primates, where distinct differences to the modern human dentition are observed [11]. The exact mechanism leading to their absence is not well understood, however, several theories have been proposed; either stressing the role of environmental factors or emphasizing the genetic origin of third molar agenesis.

Local factors that have been associated with missing third molars are delays in growth or in tooth formation [15,18], space deficiency in the dental arches [35] and certain dental interventions [36]. In addition, in cases with smaller dentoalveolar processes, there is a higher incidence of third molar agenesis [21], which might be a result of dietary differences between populations [21], or more likely a sign of variation related to sex or historical population differences [3,18]. Additionally, being the last tooth that develops in the human dentition, the third molar is more prone to presenting disturbances in its normal development, including its absence. This agrees with the inhibitory cascade mechanism, described by Evans et al. [11] and Kavanagh et al. [13], based on which, the size of the first and second molars can predict the size of the third molar. Assuming that a size reduction is a precursor of third molar agenesis, this mechanism supports the common absence of third molars in humans.

Nevertheless, there is also ample evidence of a genetic origin for third molar agenesis [5,6,37,38], related to an evolutionary mechanism leading to a gradual decrease in tooth number as well as tooth and jaw size [11,13,17,39]. According to this evolutionary theory, natural selection has led to large variation in cranial morphology among populations, including variation in the breadth of the cranium and the sagittal dimension of the midface. This is merely attributed to adaptations of the craniofacial complex to climate conditions and temperature [39]. The large variation among modern human populations in the prevalence of third molar agenesis [3] may be a result of this evolutionary mechanism and its different phenotypic expressions in various parts of the world.

In the present population, there was a clear difference in facial size between individuals with missing thirds molars and normal controls, providing support to the theory that this mechanism may continue to control the evolution of the human craniofacial complex. The largest effects were observed in the maxilla and the mandible, with size differences exceeding 3%. When the same association was evaluated in cases with non-syndromic tooth agenesis of permanent teeth other than third molars, affected individuals also presented a smaller maxillary configuration, but no effect was detected in the mandible [16]. This contrasting finding might be related to the timing of mandibular development and its correspondence to the development of third molars, especially at later developmental stages. Although the formation of all other permanent teeth chronologically precedes the peak of mandibular growth [40], this is not true for third molars, which on average start to develop after the age of 10 and are completely formed in early adulthood [41]. Thus, the mandibular structures share much more common developmental timing with the third molars than with other teeth. Therefore, we hypothesize that any association of tooth agenesis with mandibular size, at earlier developmental stages, might have been skewed by other factors, later during development, whereas this was not the case for third molar agenesis, which was clearly associated with a decreased size of the mandible. 

The above argument is strengthened by the fact that, developmentally, the mandible is a complex configuration of various modules, which stem from different mesenchymal precursors [42,43]. Although there are multiple theories about the mechanisms that control the integration of these modules, none of them manages to adequately explain the complexity of their structural relationships. This pertains particularly to the integration between muscle and tooth-bearing parts of the mandible, which remains largely equivocal [44]. The absence of third molars is an additional factor for variation in the development of the dentoalveolar process as well as its proximal anatomical parts. Phenotypically, this could lead to size variations, such as the ones observed in this study. Furthermore, it is well documented that the mandible presents high adaptability to environmental stimuli, such as chewing forces and dietary habits [45,46]. Human populations from various geographic regions exhibit distinct differences in craniofacial anatomy, linked to patterns of subsistence. Populations, for example, that followed soft agricultural diets displayed smaller crania compared to populations consuming harder foods [45]. The dietary habits of modern humans include more processed foods requiring less masticatory effort to be consumed; thus, it is plausible that the masticatory system shows adaptive changes including smaller mandibles and missing third molars. Such mechanisms have been formerly described, as driving forces of craniofacial evolution in modern humans, however, their effect is most likely much weaker compared to other factors for craniofacial variation [21,47].

In support of these assumptions, this study also showed that the number of missing third molars was negatively associated with the size of the maxilla, the mandible and the overall facial configuration, meaning that the fewer third molars were present, the smaller the skeletal sizes. In order to further examine a potential local effect of third molar agenesis on the size of the respective jaw, the sample was divided into subgroups consisting of individuals with only maxillary or only mandibular third molar agenesis, investigated against individuals without any third molar agenesis. The results showed that in cases with missing third molars only in the maxilla or only in the mandible, the effect on size was still evident in both jaws. Probably due to the small sample size of the subgroups, the statistical analysis for mandibular size did not show a significant effect in the group with isolated maxillary third molar agenesis, however, the trend was the same as in the other subgroup. This argues primarily for a biological association between third molar agenesis and jaw size, rather than a localized effect of internal or external stimuli. The finding is also in line with previously reported results, by this group, on a large sample of individuals with permanent tooth agenesis. It appears that both phenotypes are likely an expression of the same developmental mechanism controlling the relationship between tooth number, with tooth and jaw size [16]. This mechanism might be the result of an evolutionary process observed in humans, leading to fewer and smaller teeth, as well as smaller facial structures. However, this theory still needs to be evaluated and validated by further investigations.

Our findings suggest that isolated third molar agenesis is part of a developmental cascade resulting also in craniofacial size reduction. This might be the effect of an evolutionary process observed in humans, leading to less and smaller teeth, as well as smaller facial structures.

## 5. Conclusions

The present study showed a robust association between agenesis of third molars and facial size in modern humans. Individuals with one or more missing third molars present smaller maxillas, mandibles and overall facial configurations compared to individuals with all teeth present. We propose that, instead of an isolated outcome, missing third molars are most likely a phenotypic expression of a developmental mechanism also controlling craniofacial size in humans. This observation might be related to the evolutionary mechanism observed in humans, which leads to less and smaller teeth and a smaller face.

## Figures and Tables

**Figure 1 biology-10-00650-f001:**
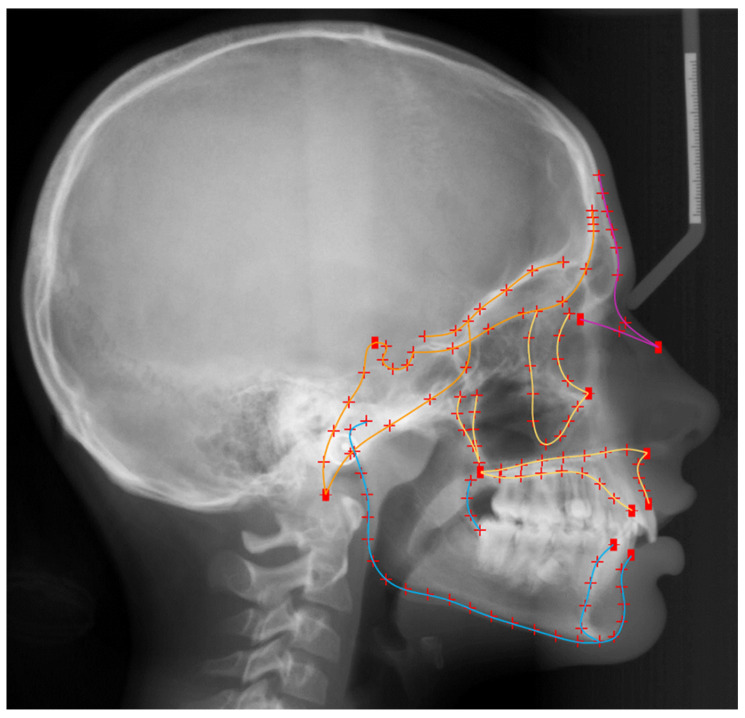
Material from: Elias S. Oeschger et al., Number of teeth is associated with facial size in humans, Scientific Reports, published 2020, Springer Nature, licensed under CC BY 4.0. Landmarks used to capture craniofacial morphology. Digitization of the craniofacial complex (n = 470) with 15 curves, which included 116 semilandmarks (red crosses), and 11 fixed landmarks (red squares). Orange color represents the structures of the cranial base, yellow the maxillary structures, blue the mandibular structures, and all lines together the whole configuration.

**Table 1 biology-10-00650-t001:** Frequency distribution of missing third molars in females, males and the entire sample.

Number of Missing Third Molars	Frequency		
	Females	Males	Total
0	186	124	310
1	23	20	43
2	27	23	50
3	9	8	17
4	31	19	50
Total	276	194	470

**Table 2 biology-10-00650-t002:** Multiple linear regression of size on age, number of missing third molars, and sex.

Size Configurations *	Parameter	β-Coefficient	95% CI		
			Lower Bound	Upper Bound	*p*-Value
Cranial Base	Intercept	4.955	4.943	4.966	<0.001
	Age	0.001	0.001	0.002	<0.001
	Number of missing third molars	−0.002	−0.004	0.001	0.276
	Female (male: reference)	−0.026	−0.033	−0.018	<0.001
Maxilla	Intercept	4.984	4.969	4.999	<0.001
	Age	0.004	0.004	0.005	<0.001
	Number of missing third molars	−0.013	−0.016	−0.009	<0.001
	Female (male: reference)	−0.025	−0.034	−0.015	<0.001
Mandible	Intercept	5.276	5.259	5.293	<0.001
	Age	0.006	0.005	0.007	<0.001
	Number of missing third molars	−0.012	−0.016	−0.008	<0.001
	Female (male: reference)	−0.037	−0.048	−0.026	<0.001
Entire facial configuration	Intercept	6.196	6.184	6.208	<0.001
	Age	0.004	0.004	0.005	<0.001
	Number of missing third molars	−0.007	−0.010	−0.004	<0.001
	Female (male: reference)	−0.033	−0.041	−0.025	<0.001

* ln(Cs).

**Table 3 biology-10-00650-t003:** Size of individual configurations and of the entire facial configuration in subjects without missing third molars (control) and in subjects with at least one missing third molar. The values are adjusted for age and number of missing third molars.

Size Configurations *		Control		Missing Third Molars		Mean Difference
		ln (Cs)	mm	ln (Cs)	mm	
Cranial base	Females	4.95	140.76	4.95	141.35	NS
	Males	4.97	144.76	4.98	145.13	NS
Maxilla	Females	5.02	151.88	4.99	147.45	−3.00%
	Males	5.05	156.41	5.02	151.36	−3.34%
Mandible	Females	5.32	204.77	5.29	198.20	−3.31%
	Males	5.36	213.59	5.32	206.15	−3.60%
Entire facial configuration	Females	6.22	504.96	6.21	497.14	−1.57%
	Males	6.26	524.20	6.24	513.99	−1.99%

* ln(Cs).

**Table 4 biology-10-00650-t004:** Multiple linear regression of size on age, number of missing third molars, and sex, performed on subsamples with no third molar agenesis or agenesis only in the maxilla or only in the mandible.

Size Configurations *	Parameter	β-Coefficient	95% CI		
			Lower Bound	Upper Bound	*p*-Value
**Individuals with third molar agenesis only in the maxilla (N = 37) and no agenesis (N = 310)**					
Maxilla	Intercept	4.957	4.940	4.973	<0.001
	Age	0.005	0.004	0.006	<0.001
	Sex	0.026	0.015	0.036	<0.001
	Number of missing third molars	−0.010	−0.020	0	0.048
Mandible	Intercept	5.231	5.213	5.249	<0.001
	Age	0.006	0.005	0.007	<0.001
	Sex	0.039	0.027	0.051	<0.001
	Number of missing third molars	−0.007	−0.018	0.004	0.219
**Individuals with third molar agenesis only in the mandible (N = 49) and no agenesis (N = 310)**					
Maxilla	Intercept	4.959	4.943	4.974	<0.001
	Age	0.004	0.003	0.005	<0.001
	Sex	0.026	0.015	0.036	<0.001
	Number of missing third molars	−0.018	−0.026	−0.011	<0.001
Mandible	Intercept	5.236	5.219	5.253	<0.001
	Age	0.006	0.005	0.007	<0.001
	Sex	0.038	0.026	0.050	<0.001
	Number of missing third molars	−0.021	−0.030	−0.013	<0.001

* ln(Cs).

**Table 5 biology-10-00650-t005:** Size of individual configurations in subjects without missing third molars (control) and subjects with third molar agenesis either in the maxilla or in the mandible.

Size Configurations	No Third Molar Agenesis (N = 310)		Missing Third Molars Only in the Maxilla (N = 37)			Missing Third Molars Only in the Mandible (N = 49)		
	ln (Cs)	mm	ln (Cs)	mm	Size Difference to Controls	ln (Cs)	mm	Size Difference to Controls
Maxillary size	5.03	152.93	5.01	149.90	−2%	4.99	146.94	−4% *
Mandibular size	5.34	208.51	5.31	202.35	−3%	5.29	198.34	−5% *

* Statistically significant difference.

## Data Availability

All data are available in the main text or the extended data. The protocols and datasets generated and/or analyzed during the current study are available from the corresponding author on reasonable request.

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
