# Peer review of "Third Molar Agenesis Is Associated with Facial Size"

_biology, 2021, doi:10.3390/biology10070650_

Round 1
Reviewer 1 Report
I’d like to start off by thanking the authors for their contribution to the field and for my opportunity to review it.
The major issue I see with this paper is the assertion that the observed changes result from an evolutionary mechanism. None of the evidence presented supports this hypothesis over the idea that third molar agenesis and the associated changes to the craniofacial complex are merely part of a developmental cascade that produces a less prognathic face when one or more third molars are absent. Cleft lip and palate produce phenotypic effects that can extend beyond the oral region, but there’s no argument to be made that this represents an evolutionary effect because it is clearly a deficiency of development. Similarly, while the developmental underpinnings of third molar agenesis may be unclear, the mere effect of other changes to the craniofacial complex being present when third molars are absent is not evidence for evolutionary changes.
I think that it is reasonable to discuss the possibility that this is an evolutionary change in the conclusion to your paper, but, as written, the authors suggest that they are presenting evidence for an evolutionary change. The evidence presented here is as indicative, if not more indicative of developmental processes responding to a perturbation than it is of an evolutionary process.
Materials and Methods
-It would be very useful to have a chart that shows the age distributions of your subjects, since your sample includes adolescents that with faces that are not fully developed. Those differences in facial shape are likely to influence your results, and it would be helpful to know whether or not this group is relatively evenly distributed within your sample.
-It’s not clear to me why curve and semi-landmark data were collected for this research. 116 landmarks collected for a study that includes only 470 subjects is a lot, and it seems like the only results that you report are centroid sizes that could be collected from the anatomically-defined landmarks alone. If you were performing specific comparisons for the shape of curves, I could understand, but this seems like a lot of data to collect to answer a question that the anatomical landmarks alone would have been sufficient for.
Results
I thought that the results that you obtained were very interesting. I would, however, like to see some sort of graphical distribution of the data. I’m very curious to see how the data are distributed and how much variability is present. Even male and female-specific dot plots showing the range of facial sizes organized by number of missing molars would be very helpful
Discussion
Again, I think the assertion that these data provide evidence for an evolutionary process is unsupported. It’s fine to suggest as a possibility, but a significant amount of information not present here (e.g. what genetic variants are associated with 3rd molar agenesis, what are the rates of 3rd molar agenesis in extant apes relative to humans, how does 3rd molar agenesis affect functional processes like bite force, what is the prevalence of 3rd molar agenesis in different populations and is it changing, etc) would be needed to assert the claim that 3rd molar agenesis in humans is an evolved trait.
Author Response
We would like to thank the reviewer for the very constructive comments that have helped us to notably improve our initial submission.
Review 1
Comments and Suggestions for Authors
I’d like to start off by thanking the authors for their contribution to the field and for my opportunity to review it.
Authors’ response: We would like to thank the reviewer for the comment.
The major issue I see with this paper is the assertion that the observed changes result from an evolutionary mechanism. None of the evidence presented supports this hypothesis over the idea that third molar agenesis and the associated changes to the craniofacial complex are merely part of a developmental cascade that produces a less prognathic face when one or more third molars are absent. Cleft lip and palate produce phenotypic effects that can extend beyond the oral region, but there’s no argument to be made that this represents an evolutionary effect because it is clearly a deficiency of development. Similarly, while the developmental underpinnings of third molar agenesis may be unclear, the mere effect of other changes to the craniofacial complex being present when third molars are absent is not evidence for evolutionary changes.
I think that it is reasonable to discuss the possibility that this is an evolutionary change in the conclusion to your paper, but, as written, the authors suggest that they are presenting evidence for an evolutionary change. The evidence presented here is as indicative, if not more indicative of developmental processes responding to a perturbation than it is of an evolutionary process.
Authors’ response: The reviewer makes a valid point regarding the interpretation of our study results. In the revised manuscript we elaborate the ambiguity in current evidence regarding third molars agenesis in humans and removed emphasis from the notion that our results are indicative of an evolutionary process. We have added related bibliography to underpin the uncertainty in the process linked to congenital absence of third molars.
Materials and Methods
-It would be very useful to have a chart that shows the age distributions of your subjects, since your sample includes adolescents that with faces that are not fully developed. Those differences in facial shape are likely to influence your results, and it would be helpful to know whether or not this group is relatively evenly distributed within your sample.
Authors’ response: Supplementary Figure 1 shows the age distribution in our sample. As reported in our results, and as expected, age had a strong effect in the size of all configurations, thus we controlled for this factor in our regression analysis.
-It’s not clear to me why curve and semi-landmark data were collected for this research. 116 landmarks collected for a study that includes only 470 subjects is a lot, and it seems like the only results that you report are centroid sizes that could be collected from the anatomically-defined landmarks alone. If you were performing specific comparisons for the shape of curves, I could understand, but this seems like a lot of data to collect to answer a question that the anatomical landmarks alone would have been sufficient for.
Authors’ response: The reviewer raises a fair point. This investigation is part of a larger investigation assessing the craniofacial structures in individuals with tooth agenesis and other phenotypic expression of dental anomalies. The present sample is, thus, part of a larger subject group, which was analyzed using the same methodology. This explains the use of more landmarks than needed for studying size.
Results
I thought that the results that you obtained were very interesting. I would, however, like to see some sort of graphical distribution of the data. I’m very curious to see how the data are distributed and how much variability is present. Even male and female-specific dot plots showing the range of facial sizes organized by number of missing molars would be very helpful
Authors’ response: Following the reviewer’s recommendation, we have added graphs to the Supplementary Material submitted with the revised version of our manuscript. Supplementary Figures 2a-d show scatterplots displaying size variability according to age, in females and males. Supplementary Figures 3a-d present box plots showing size variability organized by number of missing third molars, in females and males.
Discussion
Again, I think the assertion that these data provide evidence for an evolutionary process is unsupported. It’s fine to suggest as a possibility, but a significant amount of information not present here (e.g. what genetic variants are associated with 3rd molar agenesis, what are the rates of 3rd molar agenesis in extant apes relative to humans, how does 3rd molar agenesis affect functional processes like bite force, what is the prevalence of 3rd molar agenesis in different populations and is it changing, etc) would be needed to assert the claim that 3rd molar agenesis in humans is an evolved trait.
Authors’ response: As mentioned above, we accept the reviewer’s comment and would like to thank them for raising this point. In the revised manuscript we have adjusted our statements and included related bibliography in order to present a more accurate interpretation of our results. We think that the reviewer’s comment has helped us to significantly improve our manuscript.
Reviewer 2 Report
The manuscript of Nikolaos Gkantidis and coauthors entitled "Third molar agenesis is associated with facial size" reports interesting data on the association between agenesis and facial size. The paper is short, concise and well written. The methods are adequate, sample size on the low side but good enough for the purpose of the study.
I have only two concerns one on the reporting of the regressions as tables. Probably using coefficient plot could be more visual. In table 2 and 4 as well as L171, the authors suggest they used Multivariate regression, which is not the case. Multivariate regression is when you have several dependent (Y) variables, here there is only 1 dependent variable: ln(CS), so it is a multiple regression (several predictors). There is sometimes some confusion even in the stat literature, but as here they used geometric morphometrics (even if they don't analyze the shape per se) it should be avoid.
My second concern is about the introduction and discussion. There are developmental hypotheses in the literature about third molar agenesis (eg gradient field). The publications that the authors belong to one of these hypotheses but here they don't really express them. They don't make the link between this developmental field reduction and size reduction of tooth bearing structure but on the discussion. I think it is part the working hypothesis and could be presented a little bit in the introduction. On this note, as the sentence L 51 is built it seems to me that ref 13 suggest that agenesis is related to decrease functional needs, which is not the case.
L269. The authors suggest that there are some adaptive changes in relation to soft food. I'm not really convinced that consuming soup is a strong driver of selection (relaxing selection maybe). I'm not sure ref 10,12,13 say anything about that.
Author Response
We would like to thank the reviewer for the thoughtful comments that have helped us improve our initial submission.
Comments and Suggestions for Authors
The manuscript of Nikolaos Gkantidis and coauthors entitled "Third molar agenesis is associated with facial size" reports interesting data on the association between agenesis and facial size. The paper is short, concise and well written. The methods are adequate, sample size on the low side but good enough for the purpose of the study.
Authors’ response: We thank the reviewer for the overall evaluation of our study.
I have only two concerns one on the reporting of the regressions as tables. Probably using coefficient plot could be more visual. In table 2 and 4 as well as L171, the authors suggest they used Multivariate regression, which is not the case. Multivariate regression is when you have several dependent (Y) variables, here there is only 1 dependent variable: ln(CS), so it is a multiple regression (several predictors). There is sometimes some confusion even in the stat literature, but as here they used geometric morphometrics (even if they don't analyze the shape per se) it should be avoid.
Authors’ response: The reviewer is correct that coefficient plots would provide a more visual presentation of the results. We chose tables in order to provide the readership with a clear presentation of the values needed to execute the equation of the regression. Therefore, we would prefer to keep it as is and show our results in the form of tables. In regard to the regression analysis, the reviewer makes valid observation. We have adjusted the language and rephrased the description of our statistical analyses.
My second concern is about the introduction and discussion. There are developmental hypotheses in the literature about third molar agenesis (eg gradient field). The publications that the authors belong to one of these hypotheses but here they don't really express them. They don't make the link between this developmental field reduction and size reduction of tooth bearing structure but on the discussion. I think it is part the working hypothesis and could be presented a little bit in the introduction. On this note, as the sentence L 51 is built it seems to me that ref 13 suggest that agenesis is related to decrease functional needs, which is not the case.
Authors’ response: Following the reviewer’s suggestions we have adjusted the revised manuscript and presented the association between tooth size and the size of the dentoalveolar structures in the introduction, in more detail.
L269. The authors suggest that there are some adaptive changes in relation to soft food. I'm not really convinced that consuming soup is a strong driver of selection (relaxing selection maybe). I'm not sure ref 10,12,13 say anything about that.
Authors’ response: We included this statement based on previous studies that have described differences in the masticatory apparatus between populations with different dietary habits (see Reference 21). However, we have rephrased it in the revised discussion in order to avoid making a strong statement about a mechanism that has a relatively weak impact on craniofacial variability in humans. Furthermore, we have revised our references to better support the statements made in the manuscript. We would like to thank the reviewer for the very constructive comments.